# Sex Differences in Supplement Intake in Recreational Endurance Runners—Results from the NURMI Study (Step 2)

**DOI:** 10.3390/nu13082776

**Published:** 2021-08-13

**Authors:** Katharina Wirnitzer, Mohamad Motevalli, Derrick R. Tanous, Martina Gregori, Gerold Wirnitzer, Claus Leitzmann, Thomas Rosemann, Beat Knechtle

**Affiliations:** 1Department of Subject Didactics and Educational Research and Development, University College of Teacher Education Tyrol, 6010 Innsbruck, Austria; katharina.wirnitzer@ph-tirol.ac.at; 2Department of Sport Science, University of Innsbruck, 6020 Innsbruck, Austria; mohamad_motevali@yahoo.com (M.M.); derrick.tanous@student.uibk.ac.at (D.R.T.); 3Life and Health Science Cluster Tirol, Subcluster Health/Medicine/Psychology, 6020 Innsbruck, Austria; 4Research Center Medical Humanities, Leopold-Franzens University of Innsbruck, 6020 Innsbruck, Austria; 5Faculty of Physical Education and Sports Sciences, Ferdowsi University of Mashhad, Mashhad 9177948979, Iran; 6Department of Nutritional Sciences, University of Vienna, 1090 Vienna, Austria; a00805153@unet.univie.ac.at; 7AdventureV & Change2V, 6135 Stans, Austria; gerold@wirnitzer.at; 8Institute of Nutrition, University of Gießen, 35390 Gießen, Germany; claus@leitzmann-giessen.de; 9Institute of Primary Care, University of Zurich, 8091 Zurich, Switzerland; thomas.rosemann@usz.ch; 10Medbase St. Gallen Am Vadianplatz, 9001 St. Gallen, Switzerland

**Keywords:** gender differences, supplement, ergogenic aids, endurance, running, athletes, vegan, vegetarian, plant-based

## Abstract

It has been well-documented that female and male athletes differ in many physiological and psychological characteristics related to endurance performance. This sex-based difference appears to be associated with their nutritional demands including the patterns of supplement intake. However, there is a paucity of research addressing the sex differences in supplement intake amongst distance runners. The present study aimed to investigate and compare supplement intake between female and male distance runners (10 km, half-marathon, (ultra-)marathon) and the potential associations with diet type and race distance. A total of 317 runners participated in an online survey, and 220 distance runners (127 females and 93 males) made up the final sample after a multi-stage data clearance. Participants were also assigned to dietary (omnivorous, vegetarian, vegan) and race distance (10-km, half-marathon, marathon/ultra-marathon) subgroups. Sociodemographic characteristics and the patterns of supplement intake including type, frequency, dosage, and brands were collected using a questionnaire. One-way ANOVA and logistic regression were used for data analysis. A total of 54.3% of female runners and 47.3% male runners reported consuming supplements regularly. The frequency of supplement intake was similar between females and males (generally or across dietary and distance subgroups). There was no significant relationship for sex alone or sex interactions with diet type and race distance on supplement intake (*p* < 0.05). However, a non-significant higher intake of vitamin and mineral (but not CHO/protein) supplements was reported by vegan and vegetarian (but not by omnivorous) females compared to their male counterparts. In summary, despite the reported findings, sex could not be considered as a strong modulator of supplement intake among different groups of endurance runners.

## 1. Introduction

Sex-based differences in sports and exercise nutrition have been considered an important area of debate, particularly during the recent two decades [1,2]. It has been well-established that even the minor differences between female and male athletes in physical, physiological, and psychological characteristics could result in considerable alterations in their nutritional needs [2,3,4]. Recent scientific advancements in sex-specific differences as well as an understanding of endurance athletes’ nutritional requirements have led to a remarkable progression of female endurance runners (4%) than males (1.8%) in the world records from 1985 to 2004 [5]. While the importance of sex-specific nutritional strategies is emphasized by the literature, insufficient nutrient intake in athletes may result in detrimental effects in health, performance, and adaptations [6], particularly in female and male athletes with long-term involvement in training and racing activities [7].

The effects of sex on endurance performance can be discussed from different viewpoints. In addition to the well-recognized unfavorable effects of the menstrual cycle on women’s training routines [8,9], females possess fewer erythrocytes and have lower hemoglobin levels than males, which significantly diminishes the oxygen-carrying capacity of the blood, and consequently affects endurance performance [5,8]. Additionally, fuel utilization and muscle metabolism pathways are considerably influenced by sex, as indicated by research on endurance activities [3,10]. Compared to men, women have significantly less reliance on both liver and muscle glycogen resources and a greater capacity to use muscle lipids as fuel during endurance activities [3,11]. Moreover, as women are more likely to develop a thyroid disorder than men [12], one of the most common endocrine disorders among female athletes is hypothyroidism [13], which can cause performance-limiting problems such as fatigue and/or menstrual abnormalities [13,14]. On the other hand, male athletes are more likely to be affected by cardiovascular abnormalities [15,16] that affect performance and can also lead to irrecoverable consequences on health. While the prevalence of allergies, asthmas, and other atopic diseases have been reported to be associated with age in females and males [17], endurance runners in both sexes are more likely to face an allergy problem such as hay fever and asthma compared to general populations [18].

Sex-specific differences in endurance performance are not limited to physiological characteristics. Data indicate that male athletes possess a higher degree of self-reported mental toughness than females, which may allow them to have a superior ability to cope with stress during prolonged exercise and competition [19]. However, men are more susceptible to risky behaviors such as substance abuse and the consumption of performance-enhancing substances [20,21], but women runners are known to be more health-conscious than men [22,23]. In general, given the sex-based physical differences (e.g., height, body mass, body composition, and muscle mass) [9,24], it appears to be harder for female endurance runners than their male counterparts to improve and stabilize their body composition to have an optimal level of endurance performance [25]. Moreover, it has been reported that female runners have a higher risk of unintentional caloric imbalance to reach or keep their performance-enhancing body weight [9,26].

As a result of the aforementioned differences, male and female endurance athletes may have different macronutrient and micronutrient requirements to optimize their performance and physiological adaptations during different preparation phases. For example, women’s health consciousness is associated with their attitudes toward food choice [27] as they have been identified to consume higher amounts of fruits, vegetables, and other wholesome foods than their male counterparts [27]. These plant-based food sources are reported to have higher amounts of antioxidants and carbohydrates [28], which can benefit endurance performance [29]. The nutrient requirements in endurance runners increased along with the increasing intensity, frequency, and duration of running [30], indicating an exceptional challenge of energy deficiency in the male and female distance runners compared to their counterparts in shorter races. However, if the well-recognized dietary needs of endurance athletes could be translated to a well-planned diet, evidence indicates that even ultra-endurance trials such as an ultra-endurance mountain bike stage race can be successfully and healthily completed by female athletes eating a vegan diet [31]. Available literature regarding the nutrient needs of distance runners is not consistent in covering all sex-specific issues and remains an area of debate [32]. Evidence indicates that both female and male endurance runners may not be consuming sufficient nutrients through daily foods to support their athletic requirements [26,33,34], and this concern might be more critical for endurance athletes who follow specific kinds of diets and/or inappropriately planned regimes [35,36]. Inadequacy of nutrient intake may be due to the poor nutritional knowledge of food choices [35], avoiding gastrointestinal distress, inability to manage meal frequencies as well as competition-related causes (e.g., suppressed appetite and increased anxiety) in both sexes [37]. Therefore, paying careful attention to the optimal intake of supplemental macronutrients and micronutrients has been highly recommended to both men and women endurance athletes [38,39].

Dietary supplements have been defined as products addressing nutritional, clinical, and performance goals [40]. Dietary supplement use is a common nutritional strategy; athletes frequently consume supplements to fulfill nutrient needs and enhance physical performance and/or appearance [41]. Additionally, athletes may consume supplements for further reasons such as optimizing their adaptations, avoiding exercise- or diet-induced physical and physiological distresses, or speeding up recovery and rehabilitation from injuries [42]. Although the beneficial effects of many dietary supplements on health promotion, disease prevention, and performance enhancement remain unclear [42], it is well-established that these products considerably contribute to the nutrient requirements of athletes [43]. However, despite the recent developments in theoretical dietary recommendations for endurance athletes [42,43] and the increased availability of nutritional supplements nowadays [40], nutrition professionals are challenged to provide practical guidelines for supplement intake. It has been emphasized to male and female endurance athletes to have careful consideration for a balance between potential benefits (e.g., contribution to health and performance) and risks (e.g., adverse health effects, distraction, supplement contamination) before using dietary supplements [38]. The overconsumption of dietary supplements in order to maximize their beneficial effects has been associated with diminished health status and performance, particularly with the progression of time [44]. Furthermore, considering the high prevalence of gastrointestinal (GI) disorders among female and male endurance runners [45], caution must be warranted when consuming some supplements such as amino acids, which were shown to be associated with some GI distress, particularly diarrhea [46].

To date, sex-related differences in supplement intake have been examined in different groups of athletes. Despite the well-recognized sex-based physiological and psychological differences in endurance performance, which are strongly associated with nutritional demands, there is a paucity of research comparing supplement intake patterns (as a common nutritional behavior) between male and female endurance athletes, particularly distance runners. The limited current reports did not distinguish different dietary [38,47,48] or distance [38,49] subgroups of female and male endurance runners. Therefore, the present study aimed to compare the patterns of supplement intake in female and male high-mileage runners across different subgroups based on race distance, kind of diet, and body mass index (BMI). It was hypothesized that the supplement intake differs between male and female endurance runners.

## 2. Materials and Methods

### 2.1. Study Design and Ethical Approval

This study is a part of the NURMI (Nutrition and Running High Mileage) Study, which has been conducted in multiple steps following a cross-sectional design [50]. The methods were previously described in greater detail elsewhere [22,50,51,52]. Endurance runners in the NURMI study were mainly from German-speaking countries including Germany, Austria, and Switzerland. The study protocol was approved by the ethics board of St. Gallen, Switzerland on 6 May 2015 (EKSG 14/145) with the trial registration number ISRCTN73074080.

### 2.2. Participants and Experimental Approach

Endurance runners were contacted mainly via social media, websites of organizers of marathon events, online running communities, sports/health magazines, email lists, and personal contacts as well as via additional multi-channel recruitments. Participants completed a detailed online survey within step 2 of the NURMI study (1 February to 31 December 2015), available in German and English [52]. Study participants were provided with a written description of the procedures and gave their informed consent before completing the questionnaire. The questionnaire consisted of several parts (e.g., training and racing behavior, quality of life, physical characteristics, and dietary information) including regular supplement intake (frequency per week, type, amount, and specification for nutrients). The following inclusion criteria were necessary for successful participation in the NURMI study step 2: (1) written informed consent; (2) at least 18 years of age; (3) questionnaire step 2 completed; (4) having a BMI < 30 kg/m^2^ [53,54]; and (5) successful participation in a running event of at least a half-marathon distance within the past two years. However, a total number of 79 10-km runners who had not successfully participated in either a half-marathon or a marathon also provided accurate and useable answers with plenty of high-quality data. Therefore, to avoid an irreversible loss of these valuable datasets, 10-km runners who met inclusion criteria 1–3 were included in the study population. The longest race distance reported by participants was 160 km. 

### 2.3. Data Clearance

From the initial number of 317 endurance runners, 72 participants who did not meet the general inclusion criteria including those who did not fully answer all essential questions (e.g., sex, age, race distance, diet type) were excluded from the study. Among them, three participants with a BMI ≥ 30 kg/m^2^ were excluded from data analysis to control for a minimal health status linked to a minimum fitness level, and subsequently to further enhance the reliability of datasets. As a result of the specific exclusion criteria for the present study, an additional number of 25 runners were recognized for consuming ≤50% carbohydrates in their dietary intake, which was lower than the minimum level recommended for maintaining health-performance association [35,55,56], and were excluded from the analysis to avoid conflicting data in supplement intake [55]. As a result, 220 runners (127 women and 93 men) were considered as the final sample of the present study. For data analysis, female and male participants were assigned two further classifications according to their race distance and diet types. Race distance subgroups were 10-km, half-marathon, and (ultra-)marathon. Dietary subgroups were omnivorous diet (or Western diet, with no restriction on any food), vegetarian diet (devoid of all flesh foods but include egg and/or dairy products), and vegan diet (devoid of all foods from animal sources) [57,58]. Figure 1 shows the participants’ enrollment and classifications within the present study.

### 2.4. Measures

The participants were asked to report their regular supplement intake, in addition to food frequency (unpublished data from our laboratory based on the food frequency questionnaire of the “German Health Interview and Examination Survey for Adults (DEGS)” with friendly permission of the Robert Koch Institute, Berlin, Germany) [59,60]. Macronutrient intake of female and male endurance runners was based on the self-reported daily calories from carbohydrates, protein, and fat. It should be noted that this study focused on the health and dietary aspects of supplement intake, and supplements (which are categorized based on nutrients) are described as products consumed by athletes to supply their dietary requirements of recreational running activities including participation in running events.

The questionnaire, containing mixed-format questions, was carefully and scientifically developed by leading experts in their research areas and was focused on general patterns (particularly health and dietary aspects) of supplement intake in recreational athletes participating in running events. The following items were used to assess supplement intake: intake regularity; intake frequency; kind of supplement: carbohydrate (CHO)/protein, minerals, vitamins; brand of supplement (with the nutrient that provides the main contribution); the amount of intake; the availability of other substances, to analyze sex-based differences and the potential associations with racing distance, diet types, and BMI. As the foundation of this study was based on sex-specific (not gender) differences, it is noteworthy to indicate that the term “sex” refers to the genotype, phenotype, and anatomical characteristics of a sexually reproductive organism, which is classified as either female or male and thus, was asked from the participants, accordingly [61], while “gender” is characterized as socially constructed roles, behaviors, expressions, and identities (e.g., girls, women, boys, men, gender diverse people) [61].

### 2.5. Statistical Analysis

The statistical software R version 4.0.0 (R Foundation for Statistical Computing, Vienna, Austria) was used to perform all statistical analyses. Exploratory analysis was performed by descriptive statistics (median with second and third quartiles), mean, and standard deviation (SD)). Univariate tests were performed by Chi-square tests (χ², nominal scale) to examine the association between sex, dietary, and race distance subgroups, and Kruskal–Wallis tests (ordinal and metric scale) were approximated by using the t or F distributions or using ordinary least squares and standard errors (SE) with R². Multivariate tests were performed by logistic regression analyses (95% confidence interval (95%-CI)) (displayed as effect plots), which were used to determine the effect size of the variables (sex, diet type, race distance) on kind of supplements (macronutrients, minerals, vitamins), with significant sex differences in supplement intake across dietary and race distance subgroups displayed by one-way ANOVA. The statistical level of significance was set at *p* ≤ 0.05.

## 3. Results

From a total number of 220 endurance runners with a median age of 38.5 (IQR 18.0) years, there were 100 omnivores, 40 vegetarians, and 80 vegans based on dietary subgroup; and 79 10-km runners, 84 half marathoners, and 57 (ultra-)marathoners based on race distance subgroup. Germany, Austria, and Switzerland had the majority of endurance runners with 161, 39, and 11 participants, respectively, while nine participants were from other countries including Belgium, Brazil, Canada, Italy, Luxemburg, the Netherlands, Poland, Spain, and the UK.

Descriptive analysis showed a significant sex difference in age (F_(1, 218)_ = 4.57, *p* = 0.034), where males with a median age of 42.0 (IQR 17.0) years were older than female runners with median age of 37.0 (IQR 16.0) years. While the majority of female (83%) and male (81%) runners had a normal weight (BMI: 18.5–25), the percentage of underweight (BMI < 18.5) female runners and overweight (BMI > 25) male runners was significantly higher than their counterparts in the opposed sex (χ^2^_(2)_ = 7.86, *p* = 0.020). In addition, there was a significant sex-based difference in race distance (χ^2^_(2)_ = 19.60, *p* < 0.001) with most 10-km runners and half marathoners being female, and the majority of (ultra-)marathoners were male. Furthermore, a significant difference between males and females was observed in diet type (χ^2^_(2)_ = 8.71, *p* = 0.013), as vegan and vegetarian diets were more common in females (43% and 20%, respectively) than in males (28% and 15%, respectively). While a significant sex difference was also detected in the country of residence (χ^2^_(3)_ = 10.41, *p* = 0.015), no significant association (*p* > 0.05) was found between sex and academic qualification or marital status. Table 1 shows the descriptive characteristics of the participants.

Regular intake of a supplement was reported by 54.3% of females and 47.3% of male runners. While females and males had a similar consumption of CHO/protein supplements (19% and 20%, respectively), females reported higher consumption of vitamin (46% vs. 38%, *p* = 0.232) and mineral (36% vs. 28%, *p* = 0.195) supplements compared to males. Omnivorous male and female runners had similar rates for consumption of all three types of supplements, but vegetarian and vegan females reported higher consumption of mineral and vitamin (but not CHO/protein) supplements than vegetarian and vegan males. Table 2 shows the prevalence of supplement intake among female and male runners across different diet-, distance-, and BMI-based subgroups.

Concerning the frequency of supplement intake, 59% of supplement-consuming females and 57% of males reported using supplements regularly in their daily diet (per day). The second most common frequency of supplement intake was 1–2 times per week and was reported by 14% of females and 16% of males. The remaining female and male runners reported less than once per week (7% versus 5%), 3–4 times per week (13% versus 9%), or 5–6 times per week (6% versus 14%) as the frequency of supplement consumption. When comparing female and male runners in different subgroups, no remarkable difference between female and male runners was observed in supplement intake frequency across diet-, distance-, and BMI-based subgroups (Table 3).

Results from ANOVA and the logistic regression analysis showed that diet type had a significant effect (*p* < 0.05) on the intake of vitamin supplements (particularly vegan runners had a greater intake of vitamin supplements), while no significant association was found between supplement intake and race distance, age, or BMI (*p* > 0.05) (Figure 2). The sex interactions with diet type and race distance had no significant effects (*p* > 0.05) on any type of supplement intake (Table 4 and Table 5). No significant sex-based difference was detected in CHO/protein (*p* = 0.875), mineral (*p* = 0.166), and vitamin (*p* = 0.713) intake (Table 5).

Note: Each graphic shows the results from logistic regression analysis and displays the association between a supplement type (CHO/protein, minerals, or vitamins) and a study subgroup (age, diet, distance, or BMI), which contains two sex-specified units (females on the left and males on the right) for easier comparison. Within each unit, the estimated mean effect size of supplement intake (probability with 95%-CI, upper and lower boundaries) is shown for each subgroup category. There was no significant difference between sex and the study subgroups in any type of supplement intake. 

## 4. Discussion

The present study aimed to investigate and compare the supplement intake between female and male distance runners and the potential associations with diet type and race distance. The most important findings were (1) 54.3% of female runners and 47.3% of male runners reported consuming a supplement regularly; (2) a greater intake of vitamin and mineral (but not CHO/protein) supplements were reported by vegan and vegetarian females than their male counterparts; (3) the frequency of supplement intake was similar in females and males (in general and across dietary and distance sub-groups); (4) sex did not have any significant effect on the type of supplement intake; and (5) the sex interactions with diet type and race distance had no significant effect on any type of supplement intake. As a general result, the present hypothesis was rejected, and sex-based differences in patterns of supplement intake in distance runners should be re-investigated by future detailed studies. 

In line with the present findings, most available studies indicated no general difference between male and female athletes in the number of supplement intakes [62,63,64]. However, limited and inconsistent evidence showed a significantly greater prevalence of supplement intake in male athletes than their female counterparts [65,66], or conversely, a markedly higher prevalence of supplement intake in female athletes [48,67]. In addition to the potential effects of sociodemographic factors, such sex differences reported inconsistently by studies seem to be associated with the type of supplements or the characteristics of different kinds of sports. For instance, evidence indicates that females in track and field [48] and males in strength sports [68] use more supplements compared to their counterparts of the opposite sex. Additionally, evidence shows that despite no overall sex-based difference in the amount of supplement intake [62], male athletes were more likely to consume protein and ergogenic supplements associated with maintaining and improving muscle mass/strength, and female athletes were most likely to consume micronutrient supplements, which are typically considered health-related supplements [62,63,65,66,69]. This general sex-based difference in the type of supplement intake could be associated with either the greater training volume of male than female runners [67] or the different sex-based intrinsic causes to use dietary supplements; males reported the gain of muscle mass and strength/power as a higher priority than health or meeting dietary guidelines and females mentioned health as their first priority for consuming dietary supplements [70,71]. Consistently, the present findings indicate that female runners have an 8% higher prevalence in the consumption of both minerals and vitamins, but not of CHO/protein supplements when compared to males. Regarding the type of supplement, the present results showed that vitamin supplements had the most usage prevalence in male and female runners alike. Despite consistent data indicating the higher prevalence of multivitamins among athletes [72,73], there is some contradictory evidence demonstrating proteins [62] or carbohydrates [69] are the supplements most taken by athletes, especially male athletes [69]. While this difference could potentially be contributing to the higher number of vegan/vegetarian runners in the present study, different nutritional requirements of athletes competing in different types of sports could be considered a key factor to justify this contradiction [62]. 

The present study benefited from a novel classification of participants to specify the findings of different race distances including 10-km runners, half-marathoners, or (ultra-) marathoners. Descriptive data showed that the number of females in the 10-km and half-marathon subgroups and males in the (ultra-)marathon subgroup were significantly higher than the opposite sex. While the type of supplement intake did not differ between the distance-based subgroups, sex could also not alter this association, and no difference between female and male runners was observed in supplement intake across race distance subgroups. Although no similar study has compared different race distance subgroups of endurance runners considering patterns of supplement intake yet, it is well-documented that endurance athletes generally use supplements to a greater extent than non-endurance athletes [73,74] or endurance athletes in team sports [66]. This finding might be linked with increased nutritional requirements associated with prolonged training and competition sessions [32]. Male athletes in individual endurance sports, but not in team sports, have been shown to have a higher rate of supplement intake compared to their female counterparts [66]. Unlike the null association between race distance and type of supplement intake in the present study, a recent investigation on elite athletes has shown that compared to the athletes in other track and field disciplines, long-distance runners have a significantly higher consumption of vitamins and minerals but not macronutrient supplements [75]. However, there is insufficient evidence to scientifically approve that vitamin and mineral supplements add benefits to long-distance runners, except for the conditions such as clinically diagnosed nutrient deficiency [32] or competition-related adverse outcomes (e.g., hyponatremia) [76].

In line with the increasing number of athletes who follow plant-based diets, the vegan diet has become more acceptable for improving endurance performance [35,77]. While it has been reported that about 10% of marathoners follow plant-based diets [78], the distance runners in the present study were 36% vegans and 18% vegetarians, with a significant sex-based difference in the distribution of dietary subgroups, as the number of female runners was higher than males in the vegan and vegetarian subgroups. Consistently, evidence has indicated that females are twice as likely as males to be vegan or vegetarian in Western societies [79]. In the present study, both vegan and vegetarian females reported a non-significant but considerably greater consumption of micronutrient supplements than vegan and vegetarian males, but no sex difference in the type of supplement intake was observed in the omnivorous group. In addition to the available dietary guidelines for vegan athletes that recommend supplementary intake of multi-vitamins and multi-minerals for both male and female athletes [58,80], the higher intake of micronutrient supplements by female runners with plant-based diets appears to be connected to the generally higher level of health-consciousness amongst females than males [27]. Moreover, previous findings from the NURMI study showed that vegan runners had greater health consciousness when compared to non-vegan runners [22]. However, in a study comparing the micronutrient status of vegan, vegetarian, and omnivorous endurance runners, all dietary groups consumed equal rates of micronutrient supplements, except for vitamin B_12_, with a markedly higher intake by vegan runners [81]. Research indicates that the average consumption of daily micronutrients in vegan (but not vegetarian or omnivorous) endurance runners is strongly dependent on supplement intake [49], and this might underlie why the majority of vegan (but only half of the vegetarian and the omnivorous) endurance runners could meet nutritional recommendations for most micronutrients [49]. However, it has been well-established that micronutrient deficiency can appear with any diet type [82]. As a result, supplementation of critical micronutrients (e.g., calcium, folate, magnesium, iron, copper, vitamin D, vitamin B_12_) should be recommended for all groups of athletic populations—regardless of diet type—if the composition and planning of athletes’ daily meals are inappropriate [81]. Unlike the sex differences in the consumption of micronutrient supplements among vegan and vegetarian runners, female and male runners reported a close prevalence of CHO/protein supplement intake among all dietary groups of the present study, and this is inconsistent with the previous findings indicating the higher prevalence of macronutrient supplements in male athletes [66,67]. Irrespective of sex, however, it has been well-established that a minimal CHO-to-protein ratio of at least 4:1 should be met in endurance athletes, and the previously reported ratio for sedentary individuals of 5:1 in vegans (but not the 3:1 ratio of omnivores) nicely matches this recommendation to provide a balanced nutritional status for promoting health, performance, and recovery [35,83].

The present study showed no significant association between age and type of supplement intake in both male and female runners. However, it has been shown that age could be considered as a predictor of supplement intake in athletes [84] as senior athletes are reported to have a greater consumption of dietary supplements than their junior counterparts [84,85], possibly due to the higher level of health consciousness in senior athletes compared to young athletes [84]. In the present study, while male runners were significantly older than female runners (by five years), they reported a 7% lower dietary supplement intake than females. This finding is inconsistent with the previous reports from elite athletes [86] or marathoners [78], where males had a greater supplement intake than their female counterparts by 9% and 10%, respectively. Irrespective of sex, however, the prevalence of supplement intake in both female and male runners in the present study was lower than similar reports from elite athletes in European studies [72,73,86] and non-European [71,75] studies. A possible justification is that, unlike other studies, most participants in the present study were recreational runners, who have been shown previously to have a lower intake of dietary supplements than elite athletes [62,87]. In this regard, evidence indicates that the performance level, described as the term “professionalism”, could be a more important indicator of supplement intake than age in both sexes [62,88].

Evidence shows that BMI could also be considered as a modulator of supplement intake in athletes [65], unlike the present findings where no significant association between BMI and the type of supplement intake was observed in either male or female runners. A higher prevalence of supplement intake was previously reported by overweight athletes compared to the under-weights and normal-weights [65], which might be justified by the higher lean body mass and nutritional needs of athletes competing in strength/power sports. In the present study, remarkable sex-based differences were found in the prevalence of supplement intake in underweight and overweight subgroups (Table 2). Nevertheless, these findings in the present study appear to be hardly interpretable due to the abnormal distribution of runners in BMI groups, which is highly (82%) aligned in the normal-weight (BMI: 18.5–25 kg/m^2^) subgroup, but are also likely mediated by the exclusion of runners from the final sample with a BMI ≥ 30 kg/m^2^ as well as the significantly greater number of underweight females and overweight males compared to the opposite sex.

The present study includes some limitations that must be mentioned. This study was conducted following a cross-sectional design, and the findings were based on self-report; caution must be warranted with the interpretation of the study findings. However, it has been documented that self-reports of this type of variable are valid if they are collected immediately or shortly after an event [89]. Given the sex-based nature of the present investigation, the unbalanced distribution of female and male runners (58% vs. 42%, respectively) in the total participants and amongst different subgroups might also be considered a limitation affecting the findings. In this study, the average time between completion of the last event and completion of the questionnaire by the participants was unknown (see the fifth inclusion criteria): self-reports refer to at least one adequate running event completed within the past two years). Hence, the validity of the self-report of the current study is unknown and not applicable. Control questions were implemented in different questionnaire sections to minimize validity bias and control for inconsistent and contradictory statements. As a potential selection bias, the higher proportion of vegan/vegetarian populations in DACH countries (10–14%) compared to other Western nations might have affected the present results as 55% of participants in the present study were vegans or vegetarians, which is markedly higher than the worldwide numbers as well as for German-speaking countries. Other limitations of the present study could be the lack of biochemistry assessments of nutritional status (e.g., blood and/or urine tests), which is related to the study design as well as the impossibility of conducting statistical analysis for inconsistent and incomplete open comments provided by runners who mentioned unreliable responses, missing information, or double answers for various separate or conjugated substances. These data lacked further details to be meaningfully summarized and reported. Therefore, as previously mentioned by another study on marathoners [78], caution is advised when interpreting the estimated frequencies of supplement intake due to the mixed nature of supplement products, which typically contain additional ingredients. Despite the aforementioned limitations, the present findings contribute to adding valuable novelty to the current body of evidence by considering patterns of dietary supplement intake among long-distance runners with a critical focus on female and male runners who follow various kinds of diet. 

## 5. Conclusions

The present sex-based comparison of distance runners showed a greater prevalence of supplement intake in females than males. While there was no general difference between female and male ultra-endurance runners in the type of supplement intake, the present study found no significant association between sex and supplement intake in different subgroups (including age, BMI, diet type, race distance), despite significant differences in the distribution of males and females within the above-mentioned subgroups. However, the descriptive analysis of dietary subgroups showed that vegan and vegetarian (but not omnivorous) females had a considerable—although non-significant—higher intake of vitamin and mineral (but not CHO/protein) supplements. This finding appears to be associated with the existing dietary recommendations emphasizing the importance of supplementary intake of micronutrients by vegan and vegetarian athletes to maintain the optimized level of health and performance. Generally, it seems that “sex” is not a strong modulator of supplement intake in distinct groups of endurance runners, but this parameter should be listed beside other parameters and introduced by the present study or similar investigations of future studies. Present findings may help future investigations by design to identify the sex-specific requirements of endurance runners. However, future research with large, randomized samples of distance runners can add support in providing comparable data on sex-based patterns of supplement intakes to help meet the macronutrient and micronutrient intake guidelines, which would especially contribute to a better understanding of the use of supplement intake in female and male endurance runners. 

## Figures and Tables

**Figure 1 nutrients-13-02776-f001:**
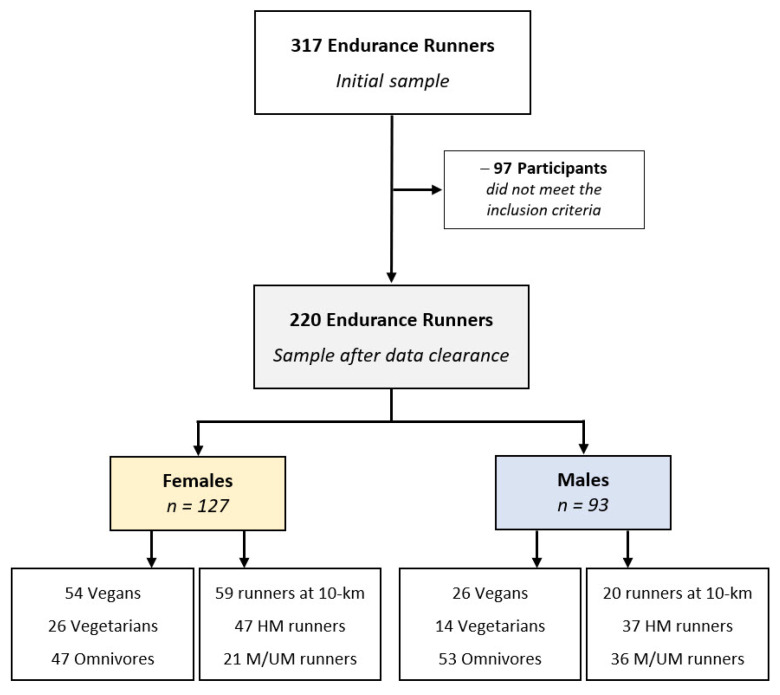
Enrollment and categorization of participants. Km—kilometers, HM—half-marathon, M/UM—marathon/ultra-marathon.

**Figure 2 nutrients-13-02776-f002:**
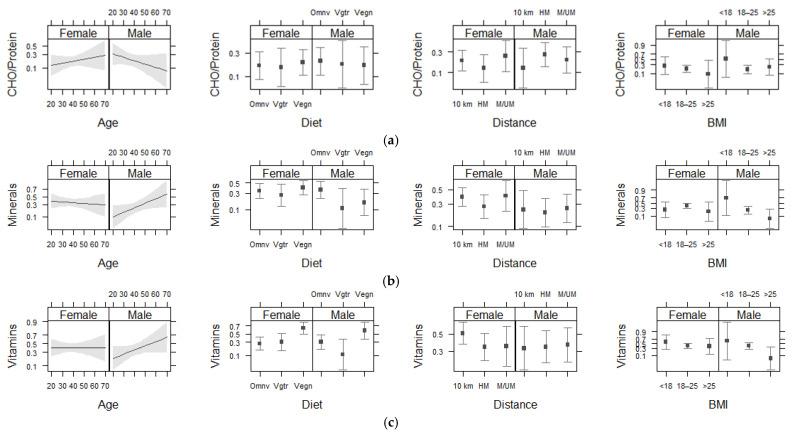
Comparison of females and males on the effects of age, diet type, race distance, and BMI on supplement intake: CHO/protein (**a**), minerals (**b**), and vitamins (**c**). CHO—carbohydrates; Omnv—omnivorous; Vgtr—vegetarian; Vegn—vegan; km—kilometers; HM—half-marathon; M /UM—marathon/ultra-marathon; BMI—body mass index. Statistical methods: Effect plots with 95% confidence intervals.

**Table 1 nutrients-13-02776-t001:** Anthropometric and sociodemographic characteristics of endurance runners.

		Total	Females	Males	Statistics
		*n* = 220	*n* = 127	*n* = 93	
Age (years)		38.5 (30, 48)	37.0 (30, 46)	42.0 (33, 50)	F_(1, 218)_ = 4.57; *p* = 0.034
Body Weight (kg)		65.0 (59, 73)	59.8 (54, 65)	73.5 (68, 80)	F_(1, 218)_ = 180.39; *p* < 0.001
Height (m)		1.70 (1.70, 1.80)	1.70 (1.60, 1.70)	1.80 (1.80, 1.80)	F_(1, 218)_ = 201.03; *p* < 0.001
BMI (kg/m^2^)	<18.5	13 (6%)	11 (9%)	2 (2%)	χ^2^_(2)_ = 7.86; *p* = 0.020
	18.5–25	181 (82%)	106 (83%)	75 (81%)	
	>25	26 (12%)	10 (8%)	16 (17%)	
Academic Qualification	No Qualification	1 (<1%)	1 (<1%)	0 (0%)	χ^2^_(4)_ = 1.96; *p* = 0.742
	Upper Secondary	72 (33%)	38 (30%)	34 (37%)	
	Education/Technical				
	A Levels or Equivalent	52 (24%)	30 (24%)	22 (24%)	
	University/Higher Degree	73 (33%)	44 (35%)	29 (31%)	
	No Answer	22 (10%)	14 (11%)	8 (9%)	
Marital Status	Divorced/Separated	11 (5%)	8 (6%)	3 (3%)	χ^2^_(2)_ = 5.55; *p* = 0.062
	Married/Living with Partner	149 (68%)	78 (61%)	71 (76%)	
	Single	60 (27%)	41 (32%)	19 (20%)	
Country of Residence	Austria	39 (18%)	14 (11%)	25 (27%)	χ^2^_(3)_ = 10.41; *p* = 0.015
	Germany	161 (73%)	100 (79%)	61 (66%)	
	Switzerland	11 (5%)	6 (5%)	5 (5%)	
	Other Countries	9 (4%)	7 (6%)	2 (2%)	
Race Distance	10-km	79 (36%)	59 (46%)	20 (22%)	χ^2^_(2)_ = 19.60; *p* < 0.001
	HM	84 (38%)	47 (37%)	37 (40%)	
	M/UM	57 (26%)	21 (17%)	36 (39%)	
Diet Type	Omnivorous	100 (45%)	47 (37%)	53 (57%)	χ^2^_(2)_ = 8.71; *p* = 0.013
	Vegetarian	40 (18%)	26 (20%)	14 (15%)	
	Vegan	80 (36%)	54 (43%)	26 (28%)	

BMI—body mass index; km—kilometers; HM—half-marathon; M/UM—marathon/ultra-marathon; Statistical methods: Kruskal–Wallis tests (F-values) and Chi-square tests (χ^2^). Note: Age, body weight, and height are presented as median (second and third quartiles).

**Table 2 nutrients-13-02776-t002:** Prevalence of supplement intake across different subgroups of female and male endurance runners.

	Supplement Intake			CHO/Protein Intake		Mineral Intake		Vitamin Intake	
	Total	Females	Males	Females	Males	Females	Males	Females	Males
Total participants	113 (51%)	69 (54%)	44 (47%)	24 (19%)	19 (20%)	46 (36%)	26 (28%)	58 (46%)	35 (38%)
(♀ = 127; ♂ = 93)									
Dietary subgroups									
Omnivorous	41 (41%)	20 (43%)	21 (40%)	9 (19%)	11 (21%)	17 (36%)	19 (36%)	14 (30%)	17 (32%)
(♀ = 47; ♂ = 53)									
Vegetarian	14 (35%)	9 (35%)	5 (36%)	4 (15%)	3 (21%)	7 (27%)	2 (14%)	8 (31%)	2 (14%)
(♀ = 26; ♂ = 14)									
Vegan	58 (72%)	40 (74%)	18 (69%)	11 (20%)	5 (19%)	22 (41%)	5 (19%)	36 (67%)	16 (62%)
(♀ = 54; ♂ = 26)									
Distance subgroups									
10-km	42 (53%)	34 (58%)	8 (40%)	12 (20%)	3 (15%)	24 (41%)	5 (25%)	32 (54%)	6 (30%)
(♀ = 59; ♂ = 20)									
HM	41 (49%)	23 (49%)	18 (49%)	6 (13%)	9 (24%)	13 (28%)	10 (27%)	18 (38%)	14 (38%)
(♀ = 47; ♂ = 37)									
M/UM	30 (53%)	12 (57%)	18 (50%)	6 (29%)	7 (19%)	9 (43%)	11 (31%)	8 (38%)	15 (42%)
(♀ = 21; ♂ = 36)									
BMI subgroups									
<18.5	8 (62%)	7 (64%)	1 (50%)	3 (27%)	1 (50%)	3 (27%)	1 (50%)	7 (64%)	1 (50%)
(♀ = 11; ♂ = 2)									
18.5–25	97 (54%)	58 (55%)	39 (52%)	20 (19%)	14 (19%)	41 (39%)	23 (31%)	46 (43%)	33 (44%)
(♀ = 106; ♂ = 75)									
>25	8 (31%)	4 (40%)	4 (25%)	1 (10%)	4 (25%)	2 (20%)	2 (12%)	5 (50%)	1 (6%)
(♀ = 10; ♂ = 16)									

CHO—carbohydrates; km—kilometers; HM—half-marathon; M/UM—marathon/ultra-marathon; BMI—body mass index; ♀—females. ♂—males; Note: Participants were allowed to choose multiple answers (more than one category of supplements). Therefore, the sum of the numbers presented in a specific row does not indicate the total number of male and female participants.

**Table 3 nutrients-13-02776-t003:** Frequency of supplement intake across different subgroups of female and male endurance runners with regular consumption of dietary supplements.

	Daily Intake		5–6 Days per Week		3–4 Days per Week		1–2 Days per Week		<1 Days per Week	
	Females	Males	Females	Males	Females	Males	Females	Males	Females	Males
Total participants	41 (59%)	25 (57%)	4 (6%)	6 (14%)	9 (13%)	4 (9%)	10 (14%)	7 (16%)	5 (7%)	2 (5%)
(♀ = 69; ♂ = 44)										
Dietary subgroups										
Omnivorous.	14 (70%)	11 (52%)	-	3 (14%)	2 (10%)	2 (10%)	2 (10%)	5 (24%)	2 (10%)	-
(♀ = 20; ♂ = 21)										
Vegetarian	4 (44%)	1 (20%)	1 (11%)	1 (20%)	2 (22%)	1 (20%)	1 (11%)	1 (20%)	1 (11%)	1 (20%)
(♀ = 9; ♂ = 5)										
Vegan	23 (57%)	13 (72%)	3 (8%)	2 (11%)	5 (12%)	1 (6%)	7 (18%)	1 (6%)	2 (5%)	1 (6%)
(♀ = 40; ♂ = 18)										
Distance subgroups										
10-km	23 (68%)	4 (50%)	2 (6%)	-	4 (12%)	2 (25%)	2 (6%)	1 (12%)	3 (9%)	1 (12%)
(♀ = 34; ♂ = 8)										
HM	10 (43%)	12 (67%)	1 (4%)	1 (6%)	3 (13%)	1 (6%)	7 (30%)	3 (17%)	2 (9%)	1 (6%)
(♀ = 23; ♂ = 18)										
M/UM	8 (67%)	9 (50%)	1 (8%)	5 (28%)	2 (17%)	1 (6%)	1 (8%)	3 (17%)	-	-
(♀ = 12; ♂ = 18)										
BMI subgroups										
<18.5	5 (71%)	1 (100%)	-	-	-	-	2 (29%)	-	-	-
(♀ = 7; ♂ = 1)										
18.5–25	32 (55%)	22 (56%)	4 (7%)	6 (15%)	9 (16%)	4 (10%)	8 (14%)	5 (13%)	5 (9%)	2 (5%)
(♀ = 58; ♂ = 39)										
>25	4 (100%)	2 (50%)	-	-	-	-	-	2 (50%)	-	-
(♀ = 4; ♂ = 4)										

km—kilometers; HM—half-marathon; M/UM—marathon/ultra-marathon; BMI—body mass index. ♀—females. ♂—males.

**Table 4 nutrients-13-02776-t004:** Sex differences in estimated mean effect size of supplement intake (probability with 95%-CI, upper and lower boundaries) across different race distance x dietary subgroups.

		Males			Females		
		CHO/Protein	Minerals	Vitamins	CHO/Protein	Minerals	Vitamins
10-km	Omnivorous	0.30 [0.15–0.52]	0.43 [0.25–0.64]	0.35 [0.18–0.56]	0.17 [0.04–0.48]	0.25 [0.08–0.55]	0.17 [0.04–0.48]
	Vegetarian	0.11 [0.02–0.50]	0.11 [0.02–0.50]	0.33 [0.11–0.67]	0.00 [0.00–1.00]	0.33 [0.04–0.85]	0.33 [0.04–0.85]
	Vegan	0.15 [0.06–0.33]	0.48 [0.30–0.66]	0.78 [0.59–0.90]	0.20 [0.03–0.69]	0.20 [0.03–0.69]	0.60 [0.20–0.90]
	Total	0.20 [0.12–0.32]	0.41 [0.29–0.54]	0.54 [0.42–0.66]	0.15 [0.05–0.38]	0.25 [0.11–0.48]	0.30 [0.14–0.53]
HM	Omnivorous	0.00 [0.00–1.00]	0.21 [0.07–0.49]	0.21 [0.07–0.49]	0.35 [0.18–0.56]	0.35 [0.18–0.56]	0.39 [0.22–0.60]
	Vegetarian	0.14 [0.04–0.43]	0.29 [0.11–0.56]	0.29 [0.11–0.56]	0.00 [0.00–1.00]	0.00 [0.00–1.00]	0.00 [0.00–1.00]
	Vegan	0.21 [0.08–0.45]	0.32 [0.15–0.55]	0.58 [0.36–0.77]	0.11 [0.02–0.50]	0.22 [0.06–0.58]	0.56 [0.25–0.82]
	Total	0.13 [0.06–0.26]	0.28 [0.17–0.42]	0.38 [0.26–0.53]	0.24 [0.13–0.41]	0.27 [0.15–0.43]	0.38 [0.24–0.54]
M/UM	Omnivorous	0.20 [0.05–0.54]	0.40 [0.16–0.70]	0.30 [0.10–0.62]	0.06 [0.01–0.31]	0.44 [0.24–0.67]	0.33 [0.16–0.57]
	Vegetarian	0.33 [0.04–0.85]	0.67 [0.15–0.96]	0.33 [0.04–0.85]	0.50 [0.17–0.83]	0.17 [0.02–0.63]	0.17 [0.02–0.63]
	Vegan	0.38 [0.13–0.72]	0.38 [0.31–0.72]	0.50 [0.20–0.80]	0.25 [0.08–0.55]	0.17 [0.04–0.48]	0.67 [0.38–0.87]
	Total	0.29 [0.13–0.51]	0.43 [0.24–0.64]	0.38 [0.20–0.60]	0.19 [0.10–0.36]	0.31 [0.18–0.47]	0.42 [0.27–0.58]
Total	Omnivorous	0.19 [0.10–0.33]	0.36 [0.24–0.51]	0.30 [0.19–0.44]	0.21 [0.12–0.34]	0.36 [0.24–0.49]	0.32 [0.21–0.46]
	Vegetarian	0.15 [0.06–0.35]	0.27 [0.13–0.47]	0.31 [0.16–0.51]	0.21 [0.07–0.49]	0.14 [0.04–0.43]	0.14 [0.04–0.43]
	Vegan	0.20 [0.12–0.33]	0.41 [0.29–0.54]	0.67 [0.53–0.78]	0.19 [0.08–0.39]	0.19 [0.08–0.39]	0.62 [0.42–0.78]
	Total	0.19 [0.13–0.27]	0.36 [0.28–0.45]	0.46 [0.37–0.54]	0.20 [0.13–0.30]	0.28 [0.20–0.38]	0.38 [0.28–0.48]

CHO—carbohydrates; km—kilometers; HM—half-marathon; M/UM—marathon/ultra-marathon; Statistical methods: Logistic regression analysis (95%-CI).

**Table 5 nutrients-13-02776-t005:** ANOVA results for effects of sex, diet, distance, and the associated interactions on supplement intake.

	CHO/Protein			Minerals			Vitamins		
	χ²	Df	*p*	χ²	Df	*p*	χ²	Df	*p*
Sex	0.02	1	0.875	1.92	1	0.166	0.14	1	0.713
Age	0.00	1	0.997	1.48	1	0.224	1.37	1	0.242
BMI	1.13	2	0.567	5.28	2	0.071	5.11	2	0.078
Diet	0.04	2	0.979	2.50	2	0.287	25.33	2	> 0.001 †
Distance	0.46	2	0.794	1.84	2	0.398	1.03	2	0.598
Sex * Age	2.86	1	0.091	3.16	1	0.076	2.08	1	0.149
Sex * BMI	1.26	2	0.533	2.58	2	0.275	3.80	2	0.149
Sex * Diet	0.96	2	0.619	2.68	2	0.262	1.28	2	0.526
Sex * Distance	2.64	2	0.268	0.82	2	0.664	1.70	2	0.426

* Interaction with variable(s). † Significant effects of diet type on vitamins intake; CHO—carbohydrates; Df—degree of freedom; *p*—*p*-value. Statistical methods: Analysis of variance (ANOVA) and logistic regression Chi-square test (χ²).

## Data Availability

The datasets generated during and/or analyzed during the current study are not publicly available but may be made available upon reasonable request. Subjects will receive a brief summary of the results of the NURMI study if desired.

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
