# Peer review of "Sex Differences in Supplement Intake in Recreational Endurance Runners—Results from the NURMI Study (Step 2)"

_nutrients, 2021, doi:10.3390/nu13082776_

Round 1

Reviewer 1 Report

The original article which I reviewed it is a good writing work with well organized methodology and valid results and good explanations in discussion. This is the reason that I had a few comments.  
  1. It is better to add “Figure 1” in lane 185, before the “Measures” section.
  2. Please, define that was used one-way Anova.
  3. I believe that in “Results” it would be more helpful to show the statistically significant differences on tables.
  4. These data… instead of this.

Author Response

Dear Editors,

Dear Reviewer,

Thank you for your consideration to the manuscript Nutrients-1321781 entitled “Sex Differences in Supplement Intake in Recreational Endurance Runners – Results from the NURMI Study (Step 2)”. We are grateful of your words of appreciations and the time and effort that you have dedicated to reviewing the manuscript and providing constructive comments, which have increased the quality of the manuscript.

Kind regards,

Katharina Wirnitzer, Beat Knechtle, and team of authors

Comments and Suggestions for Authors

The original article which I reviewed it is a good writing work with well-organized methodology and valid results and good explanations in discussion. This is the reason that I had a few comments.  

Answer: We are grateful of your kind consideration.

  • It is better to add “Figure 1” in lane 185, before the “Measures” section.

Answer: While we appreciate your attention, and agree that the best place to represent “Figure 1” is after the section “2.3. Data clearance”, it is notable that due to the size of this figure, which should be locate according to the template and page layout, we preferred to present this figure after the section “2.4. Measures”. However, if the upcoming revisions and preparations of the manuscript allowed us to follow your advice we will be happy to transfer this figure to one paragraph earlier (without affecting on the template and layouts). Thank you for your kind understanding.

  • Please, define that was used one-way Anova.

Answer: Thank you for bringing this point to our attention.

Action taken: We added the term “one-way” to “ANOVA” in both abstract and methods sections (lines 32 and 232).

  • I believe that in “Results” it would be more helpful to show the statistically significant differences on tables.

Answer: We understand and appreciate the reviewer’s concern. However, we would like to emphasize and remind on the fact that the baseline statistical significance between sex and other parameters are (already) presented in Table 1. In addition, due to the space limitations in Table 4, we decided to design Table 5 to express a summary of all sex-based statistical values and associations (between supplement intake and the study parameters) which are represented in Table 4. However, in Tables 2 and 3, data belongs to the descriptive representation of supplement intake and frequency, and therefore, no qualitative analysis was conducted for those data. Thank you for your kind understanding.

  • These data… instead of this.

Answer: Thanks for mentioning this great point.

Action taken: We revised the term accordingly (line 490).

Reviewer 2 Report

In this manuscript, the authors present the findings of an analysis of a sub-set of data collected as part of a larger study of endurance runners.

The authors present some interesting data, but overall I have several remaining questions and suggestions that should be addressed before this can be considered for publication.

  1. While the premise of this manuscript (female vs male supplement intake) is interesting, there is still limited data on supplement intake in athletes, particularly data with sample sizes larger than a single sports team. As such, this paper has the potential to explore supplement intake in endurance runners, as well as relevant associations. In some instances, these associations may not relate to sex/gender and I would suggest that the data could be analysed more robustly with the full sample set. Furthermore, given this data was collected as part of a larger study exploring differences between vegetarian and non-vegetarian athletes, it would be expected that diet type is the key variable of interest.
  2. The authors present a clear definition of ‘supplement’ in the introduction. However, the presentation of supplements as vitamin, mineral or carbohydrate/protein seems limiting and perhaps excludes some types of supplements. Was the purpose of this study to focus on supplements consumed for a specific purpose? I.e. health benefits and correction of nutrient deficiency vs performance enhancement?
  3. Given the premise of the hypothesis is sex-specific differences, it is important to be very clear about the difference between sex and gender, and to note specifically what was asked of participants in the survey. A sentence or two noting the distinction between sex and gender, and confirming what participants were asked should be included in the manuscript.
  4. Can you provide a rational for excluding those with BMI > 30? If based on health-related cut-offs, why were those with BMI < 18.5 not also excluded?
  5. Please present an example and/or more detail of the questions asked in the survey. For example, what question formats were used? Please provide specific details of the questions asked regarding supplement intake.
  6. Was data collected on the type/format of supplement – i.e. bar, gel, sports drink, protein shake, etc? Were supplements described as products consumed to address nutrient insufficiency, or to enhance performance?
  7. Was data on other supplements – i.e. creatine, caffeine, etc collected?
  8. Overall, the presentation of data appears excessive given the limited amount of data collected on and described, and the overall utility of the findings. I recommend a reduction in the number of tables, or a reduction in the amount of data presented within each table. For example, it would be sufficient to summarise the ANOVA results presented in Table 5 in a single sentence.
  9. Table 1 – please note how data is displayed for body weight and height (i.e. median (IQR))
  10. Line 214 - age is presented as mean and IQR – is this a mistake?
  11. Line 222 - what was the IQR for male and female age?
  12. Line 243-244 – you report similar consumption of vitamin and mineral supplements amongst males and females – do you have accompanying p values?
  13. Figure 2 – were there any significant differences noted?
  14. The paragraph beginning on line 418 considers BMI as a modulator of supplement intake. However, this study excluded participants with a BMI > 30, and this should be noted again at this point in the discussion as it potentially skews the results and impacts the ability to make comparisons with other studies.
  15. Did any of the analyses account for dietary intake? I.e. were supplements more likely to be consumed by those with poor intake? Or were they simply contributing to an already adequate diet?
  16. Please include SD when reporting mean

Author Response

Dear Editors,

Dear Reviewer,

Thank you for your consideration and the opportunity to revise the manuscript Nutrients-1321781 entitled “Sex Differences in Supplement Intake in Recreational Endurance Runners – Results from the NURMI Study (Step 2)”, which has led to a significant improvement of the manuscript.

In response to the valuable comments mentioned by Reviewer 2, we provided some evidence along with detailed explanations that directly address each comment. We hope our responses (in below) satisfy the existing concerns of the reviewer and editors. Changes in the manuscript have been highlighted via “track changes”.

Kind regards,

Katharina Wirnitzer, Beat Knechtle, and team of authors

Comments and Suggestions for Authors

In this manuscript, the authors present the findings of an analysis of a sub-set of data collected as part of a larger study of endurance runners.

The authors present some interesting data, but overall I have several remaining questions and suggestions that should be addressed before this can be considered for publication.

  • While the premise of this manuscript (female vs male supplement intake) is interesting, there is still limited data on supplement intake in athletes, particularly data with sample sizes larger than a single sports team. As such, this paper has the potential to explore supplement intake in endurance runners, as well as relevant associations. In some instances, these associations may not relate to sex/gender and I would suggest that the data could be analysed more robustly with the full sample set.

Answer: Thank you for expressing your concern. We agree with the reviewer that the basis of study is sex-related comparisons and data should be presented basically across this variable. However, as is evident within the tables, we would like to indicate that in the majority of analysis at sub-group levels (e.g. based on diet type, race distance, BMI, age) we expressed the full sample set beside the expression of sub-analysis. We hope we could understand the reviewer’s concern correctly, however, we would be grateful if the reviewer could provide some specific examples to clear this aspect, if other sense is considered. Thank you.

Action taken: We removed the last row of Table 5, as was not relevant to the basis of the present study. Alternatively, we added 4 new and relevant rows regarding age and BMI, and their associations with sex.

  • Furthermore, given this data was collected as part of a larger study exploring differences between vegetarian and non-vegetarian athletes, it would be expected that diet type is the key variable of interest.

Answer: We completely agree with the reviewer, and it is remarkable that as is mentioned in the method section, this study is a part of the NURMI study, in which a large group of distance runners were investigated form different aspects including diet type. We believe that while the examination of potential associations between sex and diet type in supplement intake could be considered as a novel approach among endurance runners, we accept that the unbalanced diet types in the study sample could affect the relevant findings. Therefore, this was mentioned as one of the limitation in the present study (lines 467-469,471-474, 481-485).

  • The authors present a clear definition of ‘supplement’ in the introduction. However, the presentation of supplements as vitamin, mineral or carbohydrate/protein seems limiting and perhaps excludes some types of supplements. Was the purpose of this study to focus on supplements consumed for a specific purpose? I.e. health benefits and correction of nutrient deficiency vs performance enhancement?

Answer: We appreciate greatly the reviewers concern. While we approve that the presentation of supplements as vitamin, mineral, or CHO/protein could potentially exclude some sport nutrition products, we would like to note that the purpose of the present study directed us to use the nutrient-based classification due to different reasons: (1) endurance runners in the present study were mostly recreational runners who are well-characterized to follow health- rather performance-oriented purposes; (2) endurance runners, in general, have a higher level of nutritional demands compared to other athletic populations; and (3) existing data on endurance runners indicate that they are at a higher risk of consuming insufficient nutrients to support their requirements (as shortly mentioned in lines 101-103). These three reasons were the most important justification to choose nutrient-based categorization of dietary supplements. As a result, we would like to add that while definition of specified supplements seems to be out of the purpose of the present study, it could be interesting to further expand the reasons that athletes use dietary supplements. This could help to strengthen the understanding of our purpose.

Action taken: Accordingly, we added some sentences regarding different reasons of using supplements by athletes (lines 114-117).

  • Given the premise of the hypothesis is sex-specific differences, it is important to be very clear about the difference between sex and gender, and to note specifically what was asked of participants in the survey. A sentence or two noting the distinction between sex and gender, and confirming what participants were asked should be included in the manuscript.

Answer: We appreciate your valuable comment and absolutely agree with your suggestion, thank you.

Action taken: We added two statements clearly addressing the difference between sex and gender in health studies (lines 211-216), supported by a new reference (ref. 61):

Gahagan, J.; Gray, K.; Whynacht, A. Sex and gender matter in health research: addressing health inequities in health research reporting. Int. J. Equity Health. 2015, 14, 12. DOI: 10.1186/s12939-015-0144-4.

In addition, we confirmed that the participants were asked about sex (males or females) within the aforementioned statements (line 216).

  • Can you provide a rational for excluding those with BMI > 30? If based on health-related cut-offs, why were those with BMI < 18.5 not also excluded?

Answer: We understand the reviewer’s concern. Obese runners (but not overweight runners) were excluded from the study not only for health-related reasons, but more importantly due to the fact that they could not be considered athletes, especially long-distance runners. Their major motive for running might be to achieve a body-weight reduction via distance running which not only could be markedly detrimental for their health (e.g. negative impacts on joints), but also could potentially affect their nutrient and supplement intake. On the other side, the nature of endurance running matches with BMI < 18.5 (i.e., being underweight is prevalent amongst endurance runners), and it is well-established that a lower BMI is associated with an increase in endurance performance.

Action taken: In the Method, we clarified the reason for implementation of the BMI-related exclusion criteria along with mentioning the number of participants who were excluded (lines 176-179).

  • Please present an example and/or more detail of the questions asked in the survey. For example, what question formats were used? Please provide specific details of the questions asked regarding supplement intake.

Answer: Thank you for mentioning this missing information which is necessary to be clearly addressed for readers. In this regard, we would like to add that the questionnaire was focused on general patterns (particularly health and dietary aspects) of supplement intake (e.g. type, amount, frequency, ingredients, etc.) in recreational athletes participating in running events. Mixed-format questions (e.g. single-choice with “yes”/”no”, multiple choice, estimated percentage to be filled in, free/voluntary filling short text and open comments) were used within the questionnaire. Just to mention that the experts team did not intend to give the impression of having drafted a questionnaire quickly and sloppily, all questions were carefully and scientifically developed by leading experts in their areas of research and applied settings: nutritionists, Prof. Dr. Claus Leitzmann (awarded and renowned around the word) and Prof. Dr. Markus Keller; physician specialized in sports physiology and No. 1 researcher of running studies in the World, Prof. Dr. Beat Knechtle; sports psychologist, Prof. Dr. Martin Kopp; sport scientist specialized in the interface of sports, plant-based diets and health, Dr. Katharina Wirnitzer, amongst other experts.

Action taken: We added a short statement and strengthened the relevant part with further details about the questionnaire (lines 203-206).

  • Was data collected on the type/format of supplement – i.e. bar, gel, sports drink, protein shake, etc? Were supplements described as products consumed to address nutrient insufficiency, or to enhance performance?

Answer: We appreciate your concern, and agree that there should be some information regarding the specific description of “supplement” in the present study. However, we would like to state that it was out of the aim of the present study to investigate the details regarding sports nutrition products which are mostly related to pre-/in-race/during occasions, since this study focused on everyday patterns of supplement intake where running of leisure athletes is a recreational activity and participating in running events as a part of this behavior; within this, especially for vegans, supplement intake seems to play a critical role in daily dietary requirements. Although runners provided some data regarding the form of supplements throughout open comments, we would like to indicate that we were unable to scientifically summarize these data from open questions in order to provide detailed information regarding the form of supplements, ingredients, etc. This was due to the inconsistent and incomplete nature of open comments provided by runners, which is clearly explained as a limitation (lines 467-469, 491-493) in the manuscript.

Action taken: We clarified what we could describe supplement in the present study, and added some explanations to clear this aspect (lines 198-202).

  • Was data on other supplements – i.e. creatine, caffeine, etc collected?

Answer: While we appreciate the reviewer’s concern, we kindly refer the reviewer to the answers we provided for the comments no. 2 and 6 (and even no. 5), in which we clarified that the aim of the present study directed us to classify supplements based on nutrients. We hope we could be able to express our intention clearly. However, as we mentioned above, a short supplementary statement regarding further reasons for supplement intake could be appeal for readership to increase understanding of the topic as well as to indirectly justify our intention (please see the comment 3, Action taken).

  • Overall, the presentation of data appears excessive given the limited amount of data collected on and described, and the overall utility of the findings. I recommend a reduction in the number of tables, or a reduction in the amount of data presented within each table. For example, it would be sufficient to summarise the ANOVA results presented in Table 5 in a single sentence.

Answer: We understand and appreciate the reviewer’s concern. While we found a minor part in Table 5 (the lowest row) which could be removed as it seems irrelevant to the aim and scope of the present study, we would like to specify the fact that among a large number of tables derived from the quantitative and qualitative analysis, we decided to design and present these five tables as we believe that each one demonstrate valuable information.
(i) Sociodemographic data and the baseline statistical associations between sex and other parameters are presented in Table 1.
(ii)Tables 2 and 3 highlight the sex-based differences in the prevalence and frequency of supplement intake (which are considered as the most important variables in the pattern of supplement intake) across different sub-groups.
(iii) Moreover, the most important part of our qualitative analysis is presented in Table 4, but due to the space limitations, we decided to design Table 5 to express a summary of all sex-based statistical values and associations (between supplement intake and the study parameters) which we represented in Table 4.
In general, and as far as we know, there is no limitation for number of Tables (even in number of pages) in the Nutrients guidelines, we kindly ask form the reviewer to accept this issue, and generously allow us fully present our data, thank you for your kind understanding.

  • Table 1 – please note how data is displayed for body weight and height (i.e. median (IQR))

Answer: Thank you for bringing this point to our attention.

Action taken: The footnote was revised accordingly (line 260).

  • Line 214 - age is presented as mean and IQR – is this a mistake?

Answer: We are grateful of your meticulous attention. Actually “median” is correct.

Action taken: The statement was revised accordingly (line 235).

  • Line 222 - what was the IQR for male and female age?

Answer: Thank you for mentioning this missing information.

Action taken: We added the IQR values for male and female ages, and revised the wording accordingly (lines 243-244).

  • Line 243-244 – you report similar consumption of vitamin and mineral supplements amongst males and females – do you have accompanying p values?

Answer: Thank you for expressing your concern. We would like to indicate that these data belong to the descriptive presentation of supplement intake and frequency, and therefore, no qualitative analysis was conducted for those data. However, your question is vise-versa: as similar consumption between male and female was for CHO/protein, not vitamin and mineral supplements.

Action taken: We added both the p-values accordingly (line 270).

  • Figure 2 – were there any significant differences noted?

Answer: We are thankful of noting this missing information.

Action taken: we added a statement in the footnote regarding the statistical results between sex and the study subgroups (lines 317-318). Moreover, considering the 4 subgroups presented in the Figure 2, we added the missing statistical values to the Table 5 regarding the association of sex with Age and BMI in supplement intake.

  • The paragraph beginning on line 418 considers BMI as a modulator of supplement intake. However, this study excluded participants with a BMI > 30, and this should be noted again at this point in the discussion as it potentially skews the results and impacts the ability to make comparisons with other studies.

Answer: We absolutely agree with the reviewer that the BMI-related exclusion criteria should be mentioned when discussing and interpreting the findings regarding BMI.  

Action taken: We clarified the effects of BMI-related exclusion criteria on subsequent interpretations in the relevant part in discussion (lines 464).

  • Did any of the analyses account for dietary intake? I.e. were supplements more likely to be consumed by those with poor intake? Or were they simply contributing to an already adequate diet?

Answer: We respect your concern and fully agree that nutritional status is highly connected to the different aspects in patterns of supplement intake. However, as is mentioned in the Method, the STEP 2 survey was designed to ask in greatest detail about e.g. FFQ (validated DEGS food frequency questionnaire based on self-report, with friendly permission of Robert Koch Institute; but the study design is not based on a 24-h-food recall for example, which might have been even more accurate). Therefore, as currently there are three respective reports on FFQ of runners under preparation/nearly ready for submission, we hope for your kind understanding that we will present and discuss the relevant dietary aspects in full detail in the upcoming papers. Moreover, we believe that despite the close connection between “supplement intake” and “daily food intake”, these two issues are hardly measurable and interpretable independently when combined as real foods should always be the major source for micronutrient supply. Additionally, from a statistical point of view, and due to a large number of supplement brands and inconsistent declaration of ingredients used, as well as due to a big variation of product information considering supplements (type, amount, combination vs. single nutrients, etc.), it is never easy to provide meaningful and comparable results based on the self-reported data for macro- and micro-nutrients intake. Thus, we would kindly remind that with the present study we focused on general patterns of supplement intake (e.g. type, frequency, ingredients, etc.) – in addition to FFQ – in recreational runners who have been shown (by previous publication from our lab) having a high level of health consciousness regarding exercise-induced dietary behaviors:

Wirnitzer K, Boldt P, Lechleitner C, Wirnitzer G, Leitzmann C, Rosemann T, Knechtle B. (2018). Health Status of Female and Male Vegetarian and Vegan Endurance Runners Compared to Omnivores-Results from the NURMI Study (Step 2). Nutrients. 2018 Dec 22;11(1). pii: E29. doi: 10.3390/nu11010029: https://www.mdpi.com/2072-6643/11/1/29

  • Please include SD when reporting mean

Answer: If we could understand the reviewer’s concern correctly, the only place that we used mean in the previous version of the manuscript was in line 214 (which is now line 235). However, as is mentioned in the comment #9, the median is correct and we replaced with “mean”.

Round 2

Reviewer 2 Report

Thank you for your thoughtful consideration of the comments. I feel the authors have adequately addressed all comments and the quality of the manuscript is improved.